High-resolution prediction of American red squirrel in Interior Alaska: a role model for conservation using open access data, machine learning, GIS and LIDAR

Robold Richard B. rr@robold.info
Huettmann Falk
Institute of Arctic Biology, University of Alaska—Fairbanks , Fairbanks, Alaska , United States
Schuster Richard
Electronic publication date: 2021 Sep 14
Publication date: 2021
Volume: 9
Electronic Location ID: e11830
Received 2021 Jan 18; Accepted 2021 Jun 30
Copyright: © 2021 Robold and Huettmann
Copyright year: 2021
Copyright holder: Robold and Huettmann
License: This is an open access article distributed under the terms of the Creative Commons Attribution License, which permits unrestricted use, distribution, reproduction and adaptation in any medium and for any purpose provided that it is properly attributed. For attribution, the original author(s), title, publication source (PeerJ) and either DOI or URL of the article must be cited.
License URL: https://creativecommons.org/licenses/by/4.0/

Keywords: Squirrels (Tamiasciurus hudsonicus), Middens, Geographic information system (GIS), LIDAR, Machine learning, Open access, Urban subarctic

Funding: University of Alaska Fairbanks EWHALE-Lab This work was supported by University of Alaska Fairbanks; especially the EWHALE-Lab. The funders had no role in study design, data collection and analysis, decision to publish, or preparation of the manuscript.

==============================
American red squirrels (Tamiasciurus hudsonicus) are small mammals that are abundantly distributed throughout North America. Urbanization in the Anthropocene is now a global process, and squirrels live in affected landscapes. This leads to squirrels adjusting to human developments. Not much is known about the distribution of squirrels and squirrel middens near humans, especially not in the subarctic and sub-urbanized regions. Although this species is hunted, there are no real publicly available distribution and abundance estimates nor management plans and bag limits for squirrels in Alaska or in the United States known by us, except the endangered Mt. Graham squirrel. In general, insufficient squirrel conservation research is carried out; they are underrepresented in research and its literature. To further the science-based management for such species, this study aims to generate the first digital open access workflow as a generic research template for small mammal work including the latest machine learning of open source and high-resolution LIDAR data in an Open Source Geographic Information System (QGIS) and ArcGIS. Machine learning has proven to be less modeler biased and improve accuracy of the analysis outcome, therefore it is the preferred approach. This template is designed to be rapid, simple, robust, generic and effective for being used by a global audience. As a unique showcase, here a squirrel midden survey was carried out for two years (2016 and 2017). These squirrel middens were detected in a research area of 45,5 hectares (0,455 km2) in downtown Fairbanks, interior boreal forest of Alaska, U.S. Transect distances were geo-referenced with a GPS and adjusted to the visual conditions to count all squirrel middens within the survey area. Different layers of proximity to humans and habitat characteristics were assembled using aerial imagery and LIDAR data (3D data needed for an arboreal species like the red squirrels) consisting of a 3 × 3 m resolution. The layer data was used to train a predictive distribution model for red squirrel middens with machine learning. The model showed the relative index of occurrence (RIO) in a map and identified canopy height, distance to trails, canopy density and the distance to a lake, together, as the strongest predictors for squirrel midden distribution whereas open landscape and disturbed areas are avoided. It is concluded that squirrels select for high and dense forests for middens while avoiding human disturbance. This study is able to present a machine learning template to easily and rapidly produce an accurate abundance prediction which can be used for management implications.

Introduction

American red squirrels (Tamiasciurus hudsonicus, Taxonomic Serial Number TSN.: 180166), hereafter referred to as red squirrels, are common in Central and South Alaska, in Canada and in large parts of the United States and live in an arctic or subarctic climate as well as in more temperate climate zones (Steele, 1998; Hope et al., 2016). The small-scale ecological niche of red squirrels remains poorly described and quantified for subarctic climate zones even though the macro-ecological niche and the distribution of the Tamiasciurus species has been predicted by Hope et al. (2016) and the midden-site selection of red squirrels in the Yellowstone area has recently been described by Elkins et al. (2018).

Red squirrels are considered to be primarily arboreal in coniferous, deciduous and mixed forests (Steele, 1998; Rubin, 2012). They mostly occur in elevations up to 760 m and can live in sub- and urban landscapes, if coniferous forest habitat is present (Rubin, 2012). Due to the worldwide, rapid population growth of humans in the Anthropocene (Steffen, Crutzen & McNeill, 2007; Bongaarts, 2009), animals must cope with the fact that more and more habitats are developed, transformed and occupied by humans (McKee et al., 2004; Hoekstra et al., 2004; Hanski, 2011). To be able to live within human-made and—populated areas, these animals need to have a wide phenotypic plasticity. This can be shown by the degree of habituation to all forms of human behavior and influences (Luniak, 2004). Benefitting from human structures and resources might become inevitable. Beyond natural history, such situations are therefore of wider interest to science, e.g. for mitigation of urbanization processes and impact assessments of urbanization on animal populations, along with developing science-based management solutions for wildlife conservation in urbanized landscapes (Luniak, 2004; Møller, 2009). For Alaska and Fairbanks, such work in the subarctic is ongoing already, see for instance Baltensperger et al. (2013) for ravens, Baltensperger, Morton & Huettmann (2017) for marten, Mullet et al. (2017) for noise and Jochum et al. (2014) for urban wildlife conflicts and encounters.

Despite being ubiquitous in Alaskan forests, red squirrels received little attention from wildlife managers. For example, there is neither a publicly available management plan, bag limit or budget, nor a status report to be found on the website of the Alaska Department for Fish and Game (Alaska Department of Fish & Game, 2021). In addition, the density numbers the U.S. Department of Agriculture displays on the internet are derived from over 20 years old publications from 1969 and 1994 (Sullivan, 1995) that carry little specifics for Alaska’s diverse habitats. As a global phenomenon, relevant budgets and staff assigned to similar, abundant species groups with a low conservation status are usually non-existent in governance systems. Like many other species on earth, red squirrels share the fate of being marginalized by such a governance scheme, indicated by inadequate management plans in North America, resulting in ignorance, potential decline and eventually extinction (Czech & Krausmann, 2001).

Being able to capture a species distribution and the wildlife-habitat association quantitatively and in a digital fashion presents a major way forward for descriptive and predictive work (e.g. high-precision impact studies, climate change and industrial impacts). Achieving this on a high-resolution scale will further provide great progress for detailed decision-making and mitigation cases for instance. It also represents a powerful way to find out more about implied mechanisms and processes from behavioral patterns along with helping to determine the definition and components of a species’ otherwise complex ecological niche (Guisan & Zimmermann, 2000; Barry & Elith, 2006). The reliability of those niche estimations often depends on the skill of the modeler, the amount and quality of data, the quality of the chosen model algorithm (Breiman, 2001) as well as the biologist’s skills in characterizing the species (Drew, Wiersma & Huettmann, 2011). The definition of an ecological niche itself remains somewhat diverse and has been applied in several distinct ways (Soberon & Nakamura, 2009; Cushman & Huettmann, 2010). As is widely done these days, in this study the concept of an ecological niche as an n-dimensional hypervolume, described by Hutchinson (1957), was followed (Elith et al., 2006), allowing for computational analysis platforms (Drew, Wiersma & Huettmann, 2011; Humphries, Magness & Huettmann, 2018). Species distribution modeling and predictions are often carried out all over the globe; it is even shown by Svenning, Normand & Kageyama (2008) that they can be used to investigate the past. Arguably, these tend to be macro-ecological studies like the work of Hope et al. (2016).

Still, much research relies on traditional and lower performing statistical procedures like linear and logistic regressions and parsimony to predict values and goodness of fit tests for validation (Breiman, 2001; Humphries, Magness & Huettmann, 2018). This approach may have led to several irrelevant theories and questionable scientific conclusions (Breiman, 2001; see Hochachka et al., 2007, Cushman & Huettmann, 2010 and Drew, Wiersma & Huettmann (2011) for new approaches). The problem can be that conclusions are drawn from a wrong fit and theoretical model (McArdle, 1988), but not from the underlying, more complex natural mechanisms captured in the data. This can eventually result in dubious if not even false conclusions if the model itself is a poor emulation (Breiman, 2001). With machine learning,—a non-parametric method—the outcome can be based on decision trees to produce highly accurate predictions using all the possible predictors (Breiman, 2001). This might result in an initially less interpretable and more complicated model, especially for people not trained on these works (Breiman, 2001). But when inferring from predictions, it greatly enhances the outcome and accuracy (Elith et al., 2006; Drew, Wiersma & Huettmann, 2011). Further, it is easy to implement and strongly recommended (e.g. Cushman & Huettmann, 2010; Drew, Wiersma & Huettmann, 2011; Humphries, Magness & Huettmann, 2018), which is why it was chosen for this study.

To address the complex niche concepts, this study followed a rapid assessment concept as employed by Baltensperger & Huettmann (2015) and Kandel et al. (2015) to investigate, determine and quantify factors of an ecological niche of a small mammal (rodent) in the Anthropocene and to produce a predictive GIS map for inference. The main research features of this study were the easily recognizable food storing and feeding location sites of red squirrels, the so-called “middens” (Hatt, 1929; Gurnell, 1984; Steele, 1998; Boon, Reale & Boutin, 2008). For this approach, no multi-year fieldwork had to be conducted, but robust presence field data, with a virtually complete detection rate were used, linked with high resolution environmental data layers. If successful, applying this fast template on other spatial distribution investigations would save money and speed up research processes. It also presents powerful management baseline data and could boost the gathering of knowledge about species niche modeling to progress conservation. This approach can make use of the latest digital opportunities like open-source code and LIDAR data (light detection and ranging): a powerful system using light to produce new-dimensional spatial data over land cover and height (Ohse et al. (2009) for underlying GIS model concepts). LIDAR has proven to perform well in ecological modelling for squirrels, especially since it allows for the display of habitat heterogeneity and structural features (Flaherty, Lurz & Patenaude, 2014).

Applying these techniques would especially help squirrel conservation and management issues - a species group otherwise widely overlooked and underrepresented (Koprowski & Nandini, 2008, see also table 1.4 and figure 1.5 in Thorington et al., 2012) but representing the base of the food chain and presenting a major micro-predator shaping the ecological community (Feldhamer, Thompson & Chapman, 2003). Beyond being ‘cute’, small and abundant mammals, they are not really in the focus of the public for all their habitat and conservation needs. Red squirrels virtually lack a (conservation-) management program in Alaska Department of Fish & Game (2021) and in most parts of North America, as well as any abundance, population, distribution or trend estimates (Aycrigg et al., 2015); the USDA still uses density numbers which are over 20 years old (Sullivan, 1995).

The aim of this study is to develop a rapid niche assessment template, as earlier described, for red squirrels which is cost and time effective, open access and easy to apply to improve red squirrel management. Further, this investigative study was conducted to predict and then infer red squirrel midden distribution in a digital fashion with machine learning for the first time. This fills the earlier described knowledge gap of squirrel density and habitat use in the urbanized subarctic. Therefore, it helps quantify a squirrels’ ecological niche explicit in time and space as well as provide pixel-based abundance estimates using modern computational methods. This allows for progress in the science-based conservation management for such species and their habitat and aid the conservation of species.

Study Area

Considering that tree-living squirrels in the urbanized subarctic are virtually unstudied on a small landscape-scale, and certainly not in the subarctic of Alaska anywhere or in the related peer-reviewed literature (e.g. Boon, Reale & Boutin, 2008; Chen & Koprowski, 2016; Elkins et al., 2018; Kelly & Hodges, 2020), the research area was chosen as a start and to match earlier midden cruising surveys from previous years (see “Data Collection”). The earlier surveys were an exploratory ‘dry run’ and helped to improve fieldwork and to identify middens to secure squirrel occurrences; these were carried out by Ryan Adams in 2016. The research area is located North-West of Fairbanks and belongs to the University of Alaska, Fairbanks (UAF) campus (see Fig. 1). Students, staff, public sports and recreation are now dominating this study area, making it representative for a subarctic urbanized Anthropocene. In the West, the study area is demarked by a highway and by the university’s parking lot in the east, as well as by Smith Lake in the North and the Botanical garden property in the South. The area includes an ancient boreal forest surface of about 45.5 ha (0.455 km2); the axis of the study was approximately 880 m in the x- and 850 m in the y-direction. This area is located on a ridge, and has an average elevation of 145–190 m above sea level. The center coordinates of the study area are: 459,123.9 Easting and 7,193,150.6 Northing (geographic projection datum UTM NAD 83 Zone N6). The prevalent tree species are white spruce (Picea glauca), black spruce (Picea mariana) and paper birch (Betula papyrifera). The forest stand is part of the so-called ‘potato field’ and the region was used a century ago by the Tanana Chiefs as a blessed gathering site for indigenous use and celebration (nowadays referred to as ‘Troda Yoda’).

Figure 1 Research area with trail system.

The research area is enclosed by the green line, whereas the white line shows frequently used trails.

Furthermore, the study area has by now a well-developed, maintained and well-used trail system with electric-powered lights, gravel-rock walking trails and ski grooming. It is highly used year-round for hiking, snowshoeing, olympic and recreational skiing, frisbee-golf, dog walking and several ecological surveys and studies. Some of the ski and walking trails have electric light for 24 h. These structures result into year-round frequent human presence in the campus forest. The research area contains different distinct habitats such as black and white spruce forests, mixed spruce forests, general mixed forests, deciduous forests and some open grass areas and wetlands. The age and density of plants differs widely in that area along with the canopy height, but the forest is ‘mature’ and has many old-growth trees (>100 years old). The area further contains high human influence, good accessibility and a vital but virtually unstudied squirrel population within. Other than this study, a proper and available GIS and Remote Sensing mapping is absent. Overall, we find it to be a rather good representation of urbanized landscapes in interior Alaska (see also Baltensperger et al., 2013).

Methods

Study animal and midden sites

The American red squirrel (hereafter referred to as red squirrel) is a mammal, belonging to the order of Rodentia (rodents) and the family Sciuridae (squirrels). Information about its taxonomic status and biology can be found, among others, in the following references Kramm, Maki & Glime (1975), Steele (1998), Feldhamer, Thompson & Chapman (2003), Rubin (2012), Thorington et al. (2012) and Hope et al. (2016).

Red squirrels store food (mostly cones of white and black spruce P. glauca and P. mariana) in caches, named middens, feeding frequently at these food storages (Smith, 1968b; Gurnell, 1984). At these middens, cone debris accumulates and forms well-recognized piles (Gurnell, 1984). These middens are often used for many years for feeding, and consequently these debris accumulations can grow into large and persistent piles, often many decades old (Gurnell, 1984). The actual meaning of those middens, how these places exactly look like and how they are precisely counted is not clearly defined in the literature with mutual agreement. But Gurnell (1984) lists two types of midden classification: (I) clearly defined ones that consist of one huge pile and a food cache (see Fig. 2) and (II) diffused middens, which consist of several spread piles with different caching structures (see Fig. 3). Most researchers assume that each squirrel territory contains one central, primary midden (Gurnell, 1984; Boon, Reale & Boutin, 2008) and may occasionally contain several secondary middens at low population densities (Boon, Reale & Boutin, 2008). Usually, bases of old (>100 years) black or white spruce trees or logs build the supporting structure of middens (Smith, 1968b; Gurnell, 1984). An earlier study of Smith (1968b) in Interior Alaska reported two densities: one with one squirrel per 1.6 ha (0.016 km2) assuming one primary midden per squirrel and one with one squirrel per 1.2 ha (0.012 km2) (Smith, 1968b) considering inactive and abandoned middens, resulting in a midden density of 83 middens per km2.

Figure 2 Discrete midden.

The cone debris forms one midden, no other debris accumulations are visible.

Figure 3 Diffuse midden.

Several smaller piles of cone debris are located close to each other.

The ecological niche of red squirrels in this study

The concept of an ecological niche offers a powerful quantitative approach (Cushman & Huettmann, 2010; Drew, Wiersma & Huettmann, 2011) and is used following the definition of Hutchinson (1957) as n-dimensional hypervolume. This was applied in this study providing a precise and quantitative way to determine ecological niches (Stockwell, 2006; Elith et al., 2006; Drew, Wiersma & Huettmann, 2011); it primarily focuses on the species and less on its opportunities or the community it lives in Schoener (2009). In this study, the dimensions of the ecological niche are represented by the spatial predictors (see High resolution predictors, p. 14).

Data collection

In 2016, a first exploratory survey was conducted by Ryan Adams identifying 28 midden sites within the research area (see Fig. 4). No specific protocol was followed for this opportunistic survey, and the aim was simply to maximize detections. This data is used for validating the predictions.

Figure 4 Pilot midden cruising survey in 2016 and 2017.

The conducted survey of squirrel middens in 2016 is shown in red. The green markers show middens that were still present one year later.

In a more detailed reference survey in 2017, the whole area was surveyed to get an absolute midden count; no middens were missed. To achieve up to 100% survey coverage, the area was scanned by foot using transects of varying length, orientation and distance from each other. Transects were placed according to weather conditions, forest and understory type which limited the visibility (see Fig. 5).

Figure 5 Middens and survey route within the research area.

The white line indicates the survey route taken whereas the buffer around shows the average visibility. The survey route was adjusted to visibility conditions.

Since literature lacks a clear definition of red squirrel middens, only middens were counted with a cone debris pile diameter >20 cm, a fresh debris height >1 cm and at least one ground hole (=access to food cache) within 15 cm. Figure 3 shows a typical midden site within the study area. If multiple middens occurred close to each other (<10 m), all counted as one diffuse midden (no track was kept regarding the style and size of the midden). If two spatially close middens did not belong to each other, they were counted as two. This was determined by the definition of midden boundaries, size, spread of the cone debris and connecting tracks. Midden locations were collected in the field using a GPS-device (Accuracy of +/− 5m or better) and imported into ArcGIS 10.4 by ESRI.

Data analysis and treatment

For data processing and management, Ohse et al. (2009) and Zuckerberg et al. (2012) was followed as shown in the flowchart (see Fig. 6); Open Source QGIS and ArcGIS were used for geospatial processing. This data study was not funded and used existing online data following a citizen science approach on public land with a research student J1 visa to Richard Robold between the Van Hall Larenstein University of Applied Science in Leeuwarden, Netherlands and University of Alaska Fairbanks (UAF).

Figure 6 Flow chart of the data analysis.

The flow chart represents the workflow which resulted in the distribution model. It shows starting data sets (Euclidean distances, LIDAR imagery and presence points), the analysis steps and the final compiling of the midden distribution model.

Training data

To train the model we used the ‘complete study area’ midden dataset from 2017. These points were used as presence points of squirrel middens. Additionally, 600 random points were created with the “Create Random points” tool (ArcGIS) to obtain representative pseudo-absence data points. Because all middens in the research area were detected, the absence points actually represent real absence data. The number of absence points was chosen to ensure a sufficient density of background sampling points of 17.6 sampling points per hectare. Presence and absence points were weighted in a way that resolved in an equal ratio of presences and absences as suggested by Barbet-Massin et al. (2012); see Salford Predictive Modeler (SPM) also.

High resolution predictors

Following the concepts of Ohse et al. (2009), Herrick, Huettmann & Lindgren (2013) and Baltensperger & Huettmann (2015), a machine learning-based factor analysis using decision trees was applied on the best-available environmental predictors. For the first time for red squirrels in Alaska, a high-resolution approach (3 × 3 m) was used with landscape predictors derived from publicly available geospatial data available with FGDC (Federal Geographic Data Committee) ISO-compliant metadata (in the appendix) (Alaska Division of Geological & Geophysical Surveys, 2010). It is assumed that this is an appropriate biological scale for squirrels, providing new insights. Matching the precision and accuracy of the GPS-device, the satellite images and the LIDAR resolution, the GIS-layers were produced on this 3 × 3 m resolution. Every layer is projected in a UTM NAD 1983 N6 projection with coordinates in meters. In this study, experience is provided how this data workflow makes this approach usable as a generic template for further density and abundance estimations on a small scale and for predictions of small animals’ presence/absence in a human-driven environment.

The set of predictors used consisted of the following groups documented below.

Euclidean distance

To identify specific areas of human disturbance and natural landscape division, polygons of important landscape features were digitized with the help of optical imagery from ESRI’s World Imagery service (ESRI, 2013), which is a combination of satellite and aerial imagery (pixel size up to 30 × 30 cm). Polygons showing the outline of different structures were hand traced (ArcGIS). The tool “Euclidean Distance” was then used to produce a proximity (Euclidean distance) layer to each of these following geographic features: the lake, the walking trail for dogs in the wintertime, the ski- and snowshoe trails, the highway and buildings (maps are found in the appendix). These predictors were chosen because they represent human activities and report disturbance impacts for squirrels in the research area. The Proximities allowed to assess spatial dependencies, interactions and relationships effectively (see for instance Huettmann & Diamond, 2001 and Kandel et al., 2015 for applications).

LIDAR and satellite images

While optical imagery allows for a basic survey of the landscape, further information is needed to describe its three-dimensional form for squirrel distribution. The structure of the canopy height in particular is an essential metric for arboreal species like the red squirrel (Kemp & Keith, 1970) which can be described superiorly using LIDAR data (St-Onge & Achaichia, 2001). It provides an aerial 3D-scan of an area using laser pulses. Although there are several remote sensing techniques that try to describe vertical vegetation structure, an airborne LIDAR dataset collected by the Alaska Department of Geological and Geophysical Surveys (DGGS) provided the best available, and highest spatial resolution, as well as ease-of-use for the area of interest.

DGGS provides a ‘point cloud’ (a set of data points in three-dimensional space) of LIDAR returns with each point classified by the height (‘ground’ or ‘non-ground’) and type of the return; point spacing was 2.7 feet. The vertical accuracy of the dataset was 0.251 feet (95% confidence interval) (Alaska Division of Geological & Geophysical Surveys, 2010). Provider-classified data often leaves a significant portion of vegetated points unclassified, so the unclassified and classified ‘vegetation’ points were combined to represent all vegetation. Using the final combined class, two simple structure metrics were derived. First, canopy density was described by calculating the percentage of vegetated returns in a 3 × 3 m square. Second, tree height was calculated by finding the height of the highest vegetated return in the same grid. Isolating the ground points allowed to further describe a few significant terrain variables: aspect, slope and elevation.

To sum up the used predictors:

- Euclidean distance to the lake

- Euclidean distance to the walking trail for dogs in the wintertime

- Euclidean distance to the ski- and snowshoe trails

- Euclidean distance to the highway and to the buildings

- Canopy height

- Canopy density

- Slope

- Aspect

- Elevation.

GIS data extraction

All layers were overlaid with the presence and absence points. Data was extracted from the attribute table using the “Extract Multi-values to points” tool (ArcGIS). The resulting table is the underlying data cube used for the machine learning-based distribution model and subsequent GIS modeling.

Predictive modeling with machine learning

A predictive distribution model was created using the TreeNet machine learning algorithm in the SalfordPredictiveModeler (SPM 8.0) from Salford Systems Ltd, which provides alternatives for controlling the models’ parameters. The default setting was chosen as it provides the best-known generic solutions (400 trees, terminal branch samples per node = 2, node depth = 10) (Mi et al., 2017; see subsequent citations for their performance). As shown by Craig & Huettmann (2009) and others (Ohse et al., 2009; Humphries, 2010; Drew, Wiersma & Huettmann, 2011; Humphries, Magness & Huettmann, 2018), the algorithm is common and reliable for building ecological models. These algorithms are known to be robust and accurate for regression as well as classification (Fernandez-Delgado et al., 2014). The algorithm was trained to create a density model for squirrel middens as well as a RIO (relative index of occurrence) of middens to show the most suitable habitats for squirrels, their predicted hotspots and abundances.

To obtain spatial prediction surfaces, an equally spaced point lattice grid was created within the research area, using the “Regular Point” tool (QGIS). Data from the different predictor layers were then extracted (ArcGIS). This lattice grid was ‘scored’ in SPM 8.0 using the model created from the presence/absence points. Giving every point a value between 0 and 1, representing how likely this point is to contain a midden site (0 = not likely, 1 = very likely), this grid gives a prediction of the relative index of occurrence (RIO) of middens. Probabilities are not used here because the required parametric assumptions are not met with complex predictors and because machine learning is using a different concept (trees and splitting rules; Breiman, 2001). With the “Inverse Distance Weighting (IDW)” tool (ArcGIS), an interpolation raster was generated to get a smoothed-out prediction covering the entire research area, showing the RIO of a squirrel midden in a 3x3m resolution.

The accuracy of the model is determined by the area under the curve (AUC) of the Relative Operating Characteristic (ROC), as described by Pierce & Ferrier (2000)

Validation of the model prediction

To validate the results, the independent data of 2016 was used. Due to its smaller extent, it is of high quality for presence and absence. The RIO for these field points was extracted to assess how the predictions matched the independent field data. This way it is possible to get an unbiased quantitative assessment how good the model works in the real world.

Results

The machine learning approach of Breiman (2001) was followed inferring from predictions; the prediction surfaces are regarded as the most important outcome. The internal ROC showed an accuracy greater than 83% for midden prediction, which means this model performs above a moderate accuracy. Also, data is available, with open access for further assessment and use.

A total number of 198 middens were found within the research area following the explained methods. This midden estimate assumes 100% detectability from the surveys for the study area (45 ha; 0.45 km2). This results in a density of 4.35 squirrel middens per ha (435 per km2). The RIO also allows for a binary classification of the area: occupied or not occupied by squirrels. To determine the threshold for the area occupied, a one-sided confidence interval of 95% of the values is used, resulting in a RIO cut-off value of toccupied ≥ 0.205 (see Fig. 7). The resulting subsequent midden density for this further so-called “occupied area” is 6.3 squirrel midden per ha, and the total area occupied by squirrels amounts to 29.85 ha with 188 predicted middens.

Figure 7 Binary map showing areas where squirrels are present or absent based on the RIO.

This map shows the predicted presence distribution in green which contains 95% of presence occurrences (middens).

Analyzing the distribution of the midden points and their relation to the chosen predictors, a predictive model showing the relative index of occurrence (RIO) was generated (see Fig. 8).

Figure 8 Predictive map of the predicted relative index of occurrence of squirrel middens within the research area.

The map shows the prediction of squirrel midden occurrence. The red areas have a high probability of occurrence whereas the areas in green show low to no probability of occurrence. The trails are clearly visible as well as the open area (north-east of the research area).

In Figure 8, the highest RIO is shown in red, indicating a fragmented RIO of squirrels in the study area. This suggests clusters and patches—a population structure and the likely territories—of squirrels in the study area. High RIO areas are located along the Northern boundary on the right side of the study area where the trees are relatively high and the distance from the highway is great. In addition, the areas in the very North of the research area (close to the lake and very dense canopy) as well as in the South of the research area (old-growth forest with not too many trails) show high numbers of red patches. The parts between the dark green patch (a meadow) and the center of the research area (old-growth forest with high trees) also show a relatively high-predicted midden density which means a high possibility for midden occurrence in comparison to the rest of the area. The areas very close to the highway on the West and the middle of the lower half of the research area do not show a high RIO, which might be due to a dense understory and low trees as well as due to the high disturbance.

The map (see Fig. 8) shows that the squirrels’ midden sites mostly avoid trails (green to dark green lines throughout the research area) which are clearly highly human influenced structures. It was also found that they strictly avoid treeless areas (dark green area in the Northeast) and tend to avoid the highway area (another human influence) as well as areas with high canopy density and low canopy height.

Predictor importance

Following the approach of Breiman (2001), the predictions are the most important outcome for us.

The different predictors were scored according to their importance for the prediction of midden occurrence. In a singular view, canopy height is, according to the SPM ranking metric, the most important predictor for our midden distribution (see Table 1), supporting the conclusion that squirrels need big trees.

Table 1 Score of predictor importance.

The table shows the importance of the single predictors on a scale from 100 (most important) to 0 (relative to the most important factor).

Predictor	Score	
Canopy height	100.00	
Distance to trails	73.37	
Canopy density	71.02	
Distance to lake	69.33	
Aspect	60.18	
Distance to highway	57.37	
Distance to buildings	53.06	
Distance to the dog trail	46.31	
Elevation	44.56	
Slope	42.94	

Distance to trails, canopy density and the distance to the lake are the next important factors, all of them scoring an importance rank of approximately 70%. The next three predictors: Aspect, the distance to the highway and the distance to buildings, come after a gap and score in between 50% and 60%. The three least important factors for squirrel midden distribution seem to be the distance to the dog trail, the elevation and the slope of the ground, all scoring under 50%.

It should be kept in mind that squirrels were predicted here as a multivariate package. Synergies matter here, not individual predictors, and these consist, at least, of an interaction between canopy height, distance to trails, canopy density and distance to the lake.

Partial dependence plots

The partial dependence plots show the direction of the RIO relationship for each predictor with all else kept constant. It represents the major signals in the data. The American red squirrel prefers tall trees over small ones; the plot shows that a canopy height over ca. 9m affect the RIO in a positive way. The higher the trees are the more likely a squirrel is to build a midden in that pixel (see Fig. 9).

Figure 9 Partial dependence plot for canopy height (m).

The blue line indicates the likelihood of midden occurrence based on the canopy height. Canopy below ca. A total of 9 m has a negative impact on midden occurrence and canopy above that level and vice versa.

The trail system also affects whether and where squirrels are constructing a midden. A distance greater than 10m enlarges the RIO—it makes a pixel more likely to contain squirrel middens the further it is away from trails (see Fig. 10).

Figure 10 Partial dependence plot of the distance to trails (m).

The blue line indicates the likelihood of midden occurrence based on the distance to trails. A distance below ca. A total of 10 m has a negative impact on midden occurrence and a distance above that level and vice versa until a distance of 30 m. A total of 30–50 m distance to the trails has a negative effect on midden occurrence again.

Further, the actual density of the canopy also effects the RIO. A low canopy density reduces the possibility of a midden, and the higher the canopy density gets, the more likely a pixel is to contain squirrel middens (see Fig. 11).

Figure 11 Partial dependence plot for canopy density, ranging from 0 (no trees; 0%) to 1 (very high canopy density, 100%).

The blue line indicates the likelihood of midden occurrence based on the canopy density. A canopy density below ca. 45% has a negative impact on midden occurrence and canopy density above that level vice versa, with a rising probability of occurrence for rising canopy density.

The last strong predictor affecting the RIO is the distance from the lake. The closer to the lake, the higher the possibility of a squirrel starting a midden. Further away than ca. 250 m from such wet areas, the RIO is influenced in a negative way (see Fig. 12).

Figure 12 Partial dependence plot for the distance from Smith lake (m).

The blue line indicates the likelihood of midden occurrence based on the distance to Smith lake. A distance below ca. 250 m has a positive impact on midden occurrence and a distance above that level vice versa. The closer to the lake, the higher is the probability of midden occurrence.

Taken all together, squirrels prefer high trees, greater distance from disturbance (such as humans), show a preference for dense canopy and wet land. They seem to avoid areas that have the following characteristics: small trees and/or open with a lot of disturbance.

Once more it should be kept in mind that these predictors are all acting in concert and that a reductionist/parsimonious view, that just interprets partial dependence plots individually and selectively, is not recommended (Breiman, 2001; Humphries, Magness & Huettmann, 2018). For a true ecological understanding of squirrels, a multivariate perspective is needed. The multivariate perspective shows that human impacts are the biggest predictors for squirrels overall; they avoid human impacts but live close to humans in a safe distance.

Model validation

Opposite to frequency statistic and probabilities (Guisan & Zimmermann, 2000) in machine learning work, and when based on CARTs, the RIO is not necessarily symmetric nor carries a 0.5 cut-off. The RIO-values to the midden points from the validation survey in 2016 are relatively high. They are all greater than 0.22; the mean is 0.3523 and the highest is 0.4996. Half of the values lie between 0.2948 and 0.4366. For comparison, the highest overall predicted value is 0.6425. Those high prediction values of the previous year show that the model is able to assign high values to areas that actually contain middens and therefore performs well. Ninety-three percent of all validation values are above the average prediction score of the prediction grid points, thus the model performs significantly higher than a random guess. The midden scores of the 2016 survey score very similar to the midden locations of 2017 survey, with some outliers at the lower tail which are either mis-classified middens from the survey in 2016 or middens which were abandoned in 2017. Likely this reflects midden dynamics, which offer an interesting study to work on regarding midden turn-over time (‘blinking’) and the role of individual outlier middens.

Discussion

This is the first high resolution (3 × 3 m) predictive midden distribution model of red squirrels in Alaska, in the circumpolar subarctic, in North America and in the world. Similar studies have been done by Elkins et al. (2018), modelling the habitat of red squirrels with a generalized linear mixed model and by Pereira & Itami (1991), who modeled the habitat of the Mt. Graham Squirrel (Tamiasciurus hudsonicus grahamensis) using logistic multiple regression. Nevertheless, they differ in the use of LIDAR, machine learning and the human disturbance variables and are therefore not fully comparable. This study still agrees with mostly all findings of those mentioned above, but can further predict the occurrence of squirrel middens accurately on a small scale in an urbanized habitat in the subarctic, which was previously unknown. Flaherty, Lurz & Patenaude (2014) used LIDAR data for modelling the habitat of the European red squirrel (Sciurus vulgaris) with similar outcomes and found that LIDAR-derived explanatory variables perform well for habitat modelling, which leads to the conclusion that LIDAR is a powerful remote sensing tool and should not be overlooked.

This study provides thus the first predictive model for squirrel middens on a small scale (extent of 800 m, and 3 × 3 m resolution) in an urbanized area and in the subarctic using open access data and machine learning. Beyond squirrels, it is assumed that this model concept can break some new ground for conservation worldwide. The model gives new quantified insight explicit in time and space on the ecological niche of the red squirrel and its habitat; it allows for transparent and repeatable science as well as for the first quantitative estimations about predictors of squirrel midden distribution using open access data. Red squirrels prefer high canopy trees to build their middens (Fig. 9), which agrees with what the literature elaborates about squirrel habitat choice for the red squirrel (Kemp & Keith, 1970) as well as for the European red squirrel (Rima et al., 2010; Flaherty, Lurz & Patenaude, 2014). The fact that squirrels give preference to dense (Gurnell, 1984) or interlocking canopy to increase their foraging success and their escape possibilities from predators (Elkins et al., 2018; Steele, 1998; Vahle & Patton, 1983; Smith, 1968a) supports these outcomes.

Because squirrels avoid disturbed areas and predators in a similar way (Gill, Sutherland & Watkinson, 1996) it was expected that red squirrels also avoid human disturbance, represented through the trails (see Fig. 1). Squirrels will not build their middens right on the trail (because of humans walking right through and the lack of trees), but it is noteworthy that they try to avoid the trails until a certain ‘safety’ distance; the further away from the trail, the higher the RIO for a midden (see Fig. 10). It is also noticeable that the RIO of a midden increases with the proximity to the lake.

According to Gurnell (1984), a squirrel has approximately one midden within its territory. Considering the 198 middens counted in the research area (45.5 ha; 0.455 km2), the squirrels’ territories would have an average size of 0.23 ha (0.15 for the estimated occupied area) (2,300 m2) which is in line with reported average sizes of a squirrel’s territory of 0.24–0.98 ha (2,400–9,800 m2) (Hatt, 1929; Steele, 1998). In contrast, Smith (1968a) reports bigger territories in Interior Alaska, exceeding from 1.2 ha up to 4.8 ha (12,000–48,000 m2). It is currently unclear how middens and squirrel territories are related. Steele (1998) reports that squirrels have one or a few central middens, as does Boon, Reale & Boutin (2008) who mention one central and several secondary middens, which may occur at low population densities. Therefore, it is very difficult to estimate a population density based on the number of middens per se. For further research, it is recommended to score midden characteristics to find relationships between certain midden features, squirrel abundance and behavior. To prevent future flaws in scientific research and subsequently improve management, the use and meaning of “midden” as a definite scientific term and concept needs to be refined and finally unified.

In this study, territory sizes ranged between 0.23 ha (one midden per squirrel) and 0.92 ha (one primary and up to three secondary middens), resulting in an estimation of 49 and 198 squirrels within the study area respectively. If only the predicted occupied area is used (which contains 95% of the middens), the estimate lies between 47–188 squirrels in this area. Densities are 1.08–4.35 squirrels per ha (108–435 squirrels per km2) for the research area and 1.57–6.29 squirrels per ha (157–629 squirrels per km2) in the predicted occupied area respectively, which is similar to a found squirrel density of 1.83 squirrels per ha (183 per km2) in Interior Alaska by Brink (1964). On the other hand, compared with a reported density of 0.2–0.6 squirrels per ha (20–60 per km2) by Smith (1968b) it seems high; nevertheless, these results were found after one and two years of crop failure in white spruce. There are no current density numbers of red squirrels in Alaska, but research at Kluane Lake by Villette, Krebs & Jung, 2017 reports a squirrel density of 2.19–4.99 squirrels per ha (219–499 squirrels/km2) which is similar to the findings in this study. Therefore, the found results seem to be accurate.

Regarding future research, it would be relevant for an effective conservation management to carry out the same study on a larger scale (“scaling up” e.g. Tanana Forest in interior Alaska) to see a change with increasing distance from human structures and disturbance. It is expected that the squirrel density is increasing with increasing distance to human disturbance but simultaneously slightly decreasing through predation pressure in greater distances. This is because predators seem to keep greater spacing between themselves and humans and might limit the squirrel density in areas much further away from human disturbance. This phenomenon of indirect effects of humans on predator-prey interactions was described by Muhly et al. (2011) for large mammalian wildlife species and it would be of interest if the same is applicable for small mammals like squirrels.

Although squirrels are hunted in Alaska, there is a knowledge gap regarding abundance, distribution, population trends, viable harvest rates and sensitivities. Currently a laissez-faire approach is employed for abundant small mammal groups such as red squirrels in the US (Steiner & Huettmann, 2021), indicated by the missing management plans on the website of the Department for Fish and Wildlife in many states. A similar knowledge gap applies worldwide, especially in the tropics where many squirrel species occur but are nearly unstudied (Koprowski & Nandini, 2008). In the year 2021 this should be dramatically improved towards good and defendable professional and ecological research standards, e.g. as stated in Silvy (2012) for good wildlife management concepts and best professional practice. Game species should be managed with science-based models and research data.

Management Implications

In the spirit of rapid assessment and its global applicability (Ohse et al. (2009) and Kandel et al. (2015) for an example), the design of this study was kept as simple and cheap as possible to guarantee an easy reproducibility and transferability of this workflow for future applications. Achieving the aim of this study, a first template of how to use machine learning to produce accurate small-scale predictions for conservation and ecological management is provided. Using the same workflow, this can easily be transferred to other animals which regularly use permanent structures, such as beavers (Graf et al., 2016), prairie dogs and burrowing owls (Alverson & Dinsmore, 2014). Modelling the abundance of red squirrel middens proved how specific driving factors like canopy height and the distance to trails determine squirrel distribution in their realized niche. These important factors should subsequently be targeted in real-world management to deal with the rapid alterations of ecological niches due to climate change and anthropogenic development. This is especially important for the subarctic, where temperature will increase by about 4–9 °C by the year 2100 (IPCC, 2013), which will bring significant changes in ecosystem dynamics (Peñuelas & Filella, 2001).

Supplemental Information

Supplemental Information 1 Data values for presence and generated absence points.

The extracted local values from the predictor layers for each presence and absence point.

Click here for additional data file.

Supplemental Information 2 Data values for the lattice grid.

The extracted local values from the predictor layers for each point of the lattice grid later used to calculate the RIO-prediction.

Click here for additional data file.

Supplemental Information 3 Prediction values for the lattice grid.

The calculated RIO prediction for each point of the lattice grid. PROB_0 shows the probability for “no midden occurring”; PROB_1 shows the probability for the opposite event.

Click here for additional data file.

Supplemental Information 4 Aspect of the ground in the research area.

The aspect of the local aspect of the ground. This layer was used to extract values for the prediction.

Click here for additional data file.

Supplemental Information 5 Altitude in the research area.

The altitude within the research area. This layer was used to extract the values of the local altitude for the prediction.

Click here for additional data file.

Supplemental Information 6 Absence point distribution over the research area.

The location of the randomly generated absence points within the research area.

Click here for additional data file.

Supplemental Information 7 Binary map showing areas where squirrels are present or absent based on the RIO.

The data background to the binary map figure. The presence part contains 95% of all presence points.

Click here for additional data file.

Supplemental Information 8 Buildings within the research area.

The buildings within the research area. This layer was used to extract the values of the local distance to buildings for the prediction.

Click here for additional data file.

Supplemental Information 9 Dogtrails within the research area.

Frequently used dog trails within the research area. This layer was used to extract the values of the local distance to the dogtrails for the prediction.

Click here for additional data file.

Supplemental Information 10 Canopy density within the research area.

The canopy density within the research area. This layer was used to extract the values of the local canopy density for the prediction.

Click here for additional data file.

Supplemental Information 11 Lattice grid used to extract layer values.

This lattice grid was used to extract local values of all predictor layers to calculate the prediction.

Click here for additional data file.

Supplemental Information 12 Highway next to the research area.

The highway next to the research area. This layer was used to extract the values of the local distance to the highway for the prediction.

Click here for additional data file.

Supplemental Information 13 Canopy height within the research area.

Canopy height within the research area. This layer was used to extract the values of the local canopy height for the prediction.

Click here for additional data file.

Supplemental Information 14 Prediction calculated for the lattice grid.

The calculated prediction for each point of the lattice grid base. This layer was used to interpolate and with that generate a complete prediction for the relative occurrence of squirrel middens.

Click here for additional data file.

Supplemental Information 15 Midden locations collected in 2016.

The midden locations collected for a university intern project in 2016 were used to validate the prediction. This shapefile contains the locations of all middens found in 2016.

Click here for additional data file.

Supplemental Information 16 Boxplot of validation values, 2016.

The RIO values of the squirrel midden presence points of the cruising survey in 2016.

Click here for additional data file.

Supplemental Information 17 All middens found within the research area.

All the locations were middens that could be found within the research area during the survey for this project.

Click here for additional data file.

Supplemental Information 18 The borders of the research area.

Click here for additional data file.

Supplemental Information 19 Frequency of relative index of occurrence: presence and absence points.

The histogram shows a histogram of presence values (green) compared to absence values (grey; divided by 3, because three times more absence values were used). It shows no clear separation but a strong indication of higher values for the presence points.

Click here for additional data file.

Supplemental Information 20 Survey route for data collection.

The route that was used to survey the area and collect all the midden locations.

Click here for additional data file.

Supplemental Information 21 Frequently used trails in the research area.

The trails within the research area that are frequently used by humans for recreational purposes. This layer was used to extract the values of the local distance to the frequently used trails for the prediction.

Click here for additional data file.

Supplemental Information 22 Interpolation of the RIO-prediction lattice grid.

The complete prediction, the RIO, for squirrel middens throughout the research area. It was created from the lattice3_score using a IDW-tool.

Click here for additional data file.

Supplemental Information 23 Lake next to the research area.

The lake next to the research area. This layer was used to extract the values of the local distance to the lake for the prediction.

Click here for additional data file.

Supplemental Information 24 Slope of the ground within the research area.

The slope of the ground within the research area. This layer was used to extract the values of the local ground slope for the prediction.

Click here for additional data file.

Supplemental Information 25 The titles of each shape file.

Click here for additional data file.

Supplemental Information 26 Metadata for used LIDAR-data.

All metadata contents provided by the Alaska Division for Geographic and Geophysical Surveys for the used LIDAR dataset.

Click here for additional data file.

We especially thank Aidan Myers for preparing all the LIDAR data and Ryan Adams for the control collection in 2016. We acknowledge the University of Fairbanks, Salford Systems Ltd for their software use, IAB and the EWHALE lab and its students. We also thank the Van Hall Larenstein University of Applied Science in Leeuwarden, Netherlands and especially Mr. van Wijk. Also, thanks go out to the International Office of the UAF and a personal thanks to Mr. Jan Geisler and Mr. D. Cambu. This is EWHALE lab publication # 260.

Additional Information and Declarations

Competing Interests

Author Contributions

Data Availability

The authors declare that they have no competing interests.

Richard B Robold conceived and designed the experiments, performed the experiments, analyzed the data, prepared figures and/or tables, authored and reviewed drafts of the paper, and approved the final draft.

Falk Huettmann conceived and designed the experiments, analyzed the data, authored and reviewed drafts of the paper, and approved the final draft.

The following information was supplied regarding data availability:

The data are available in the Supplemental Files.

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
