# Peer review of "High-resolution prediction of American red squirrel in Interior Alaska: a role model for conservation using open access data, machine learning, GIS and LIDAR"

_PeerJ, doi:10.7717/peerj.11830_

## Round 0.1 · original submission · Major Revisions

We have now received two reviews that both agree this study has merit, but both suggest changes or improvements in places. Please work through the reviewer comments carefully and address them in a revised version of your manuscript.

Reviewer 1 ·

Basic reporting

Clear and unambiguous, professional English used throughout.
In general, the writing style is clear. However, there are several inaccuracies in the use of the English language throughout the manuscript which make it difficult to understand sometimes. I suggest having the manuscript proofread by someone with a high level of skill in written English.

Literature references, sufficient field background/context provided.
I suggest the authors include in the Introduction some background on previous studies that have used LiDAR and GIS for assessing squirrel habitat preferences (please see #1 y #2 in the General comments section)
The introduction should also include some background on the use of machine learning (ideally for similar studies) and provide a justification for the use of this tool in this project (i.e. what are the advantages of using machine learning for this study?)

The manuscript is well structured and conforms to PeerJ standards.
In general, figures are relevant, high quality and well labelled & described. Figure 1 should include a map of the study area in the context of Alaska (for international readers) Figures 13 and 14 can be removed as they provide little or no information and are not visually appealing

Some GIS files were supplied. However, only raster files can be loaded into QGIS. Vector files (i.e. shp) could not be loaded as only part of the file was provided. A shapefile consists of several files, of which .shp, .shx and .dbf are mandatory. These should be located in the same folder or in a zipped container. Furthermore, apart from the file name, there is no way to know what each file is. The folder should contain a metadata file (i.e. readme.txt) containing a description of each of the files provided

Experimental design

I consider the original primary research to be within the scope of the journal.
The research question is relatively well defined but it is not clearly stated how the research fills an identified knowledge gap. Thus, it is difficult to judge whether or not the research question is relevant & meaningful. Authors should clearly state in the introduction what the research question is and why this is important (i.e. what’s the knowledge gap it fills). Research aim and objectives should also be clearly stated at the end of the introduction and then discussed (in the Discussion) in the context of the study results and findings.

The Methods are well described and all procedures seem to have been performed in a rigorous way. However, the accuracy of the LiDAR derived variables (tree´s height and canopy density mainly) has not even been mentioned or discussed in the manuscript. Is this information available? If not, can the authors explain how this issue (i.e. potential inaccuracies of LiDAR derived variables) could affect the results of the study? There are also a few more issues the authors should address (please see the notes and comments in the PDF file)

Validity of the findings

Although the study findings are very interesting and well explained they should be linked -in the Discussion- with the research question and with the knowledge gap this research question attempts to fill. Study findings should also be compared with other similar studies (please see the notes and comments in the PDF file)

Additional comments

Important: I have attached an annotated PDF where I have highlighted parts of the manuscript- each highlighted part has an annotation. Please refer to the Editor if you are unable to see the annotation.

The study is interesting and the authors have done a lot a good work. However, in addition to all the comments already provided in the previous sections and in the annotated attached PDF file, authors should do a thorough search of current literature. This would greatly improve your work by providing support to some of the statements in the manuscript (please see annotated PDF) and also it would also help to better identify the contribution of your research to the exiting knowledge.
I am providing some examples of literature you could include in your article:

#1 S.Flaherty, P.Lurz and G. Patenaude, “Use of LiDAR in the conservation management of the endangered red squirrel (Sciurus vulgaris L.)” journal of Applied Remote Sensing , Vol. 8, 2014
#2 R. Nelson, C. Keller, and M. Ratnaswamy, “Locating and estimating the extent of Delmarva fox squirrel habitat using an airborne LiDAR profiler,” Remote Sens. Environ. 96(34), 292–301 (2005)
#3 Elkins, Eric & Tyers, Daniel & Frisina, Michael & Rossi, Joao & Sowell, Bok. (2018). Red Squirrel (Tamiasciurus hudsonicus) Midden Site Selection and Conifer Species Composition. Environmental Management and Sustainable Development. 7. 15. 10.5296/emsd.v7i2.12674.
#4 Jennifer K. Frey – 2003 - Initiation of Red Squirrel (Tamiasciurus hudsonicus) Monitoring on Carson National Forest, New Mexico, USDA Forest Service (Report) https://www.fs.usda.gov/Internet/FSE_DOCUMENTS/fsbdev7_011893.pdf
#5 Selecting pseudo-absences for species distribution models: how, where and how many? (2012) by Morgane Barbet-Massin et al
#6 S. Flaherty et al., The impact of forest-stand structure on red squirrel habitat use, Forestry:Int. J. For. Res. 85(3), 437444 (2012)

Annotated reviews are not available for download in order to protect the identity of reviewers who chose to remain anonymous.

Reviewer 2 ·

Basic reporting

I have provided all comments below

Experimental design

I have provided all comments below

Validity of the findings

I have provided all comments below

Additional comments

PEERJ-47121

In their manuscript titled ”High-resolution prediction of American red squirrel in Interior Alaska: A role model for conservation using open access data machine learning, GIS, and LIDAR” the authors developed a methodology for assessing land-use by red squirrels based on midden surveys and incorporating multiple ecological and anthropogenic variables associated with urban environments. The concept is of interest and I don’t have any substantive comments regarding the detailed methods or results. However, this manuscript was difficult to read due to the writing style, and a wealth of presumptive statements about the state of research on this species and small mammals in general. It was not evident to me that the authors actually know the existing literature on squirrel monitoring and management. I hope that this kind of methodological platform becomes better developed and employed for more effective management but the manuscript as it is needs significant adjustment. I have provided some associated comments below.

L22 - adaptation is an evolutionary process

L27 - please rewrite this more scientifically? The sentence “Not much small mammal research is carried out.” is anomalous, and simply not true. This needs to be put into context. Even considering only red squirrels, there is an abundance of literature and published reports on red squirrel midden survey techniques (see work by Frey, JK) as well as surveys of abundance and characteristics of middens related to squirrel densities, age structure, associated communities, 20+ years of red squirrel research including aspects of midden dynamics in Kluane, and so on.

L50 - either the authors should remove “(excluding Hawaii)” or preferably should be more explicit about where in the US red squirrels occur. There are many areas of North America besides Hawaii where red squirrels do not occur. Also, it should be noted that although the TSN for red squirrels has not yet been updated, the taxonomy for these squirrels has changed. Both Steiner and Huettmann (2021) and the current authors have failed to recognize the updated systematic relationships among North American red squirrels. See for example Illustrated Checklist of the Mammals of the World (C. J. Burgin, D. E. Wilson, R. A. Mittermeier, A. B. Rylands, T. E. Lacher, and W. Sechrest, eds.). Lynx Ediciones, Barcelona.

L51 - again, this is a myopic statement. Red squirrels are an indicator species chosen by the USDA Forest Service and for which extensive research and management has been conducted.

L52 - macro-ecological niche predictions based on WorldClim datasets were conducted by Hope et al. 2016 (MPE).

L55 - red squirrels can occur at elevations exceeding 3500m in some areas of their range. Authors must be more specific.

L62 - again, adaptation strictly adheres to selective evolutionary processes. plasticity would be more appropriate

L104-110 - again, I do not dispute that small mammals deserve much greater emphasis for conservation and management, but this sweeping broad-brush series of statements is not a rigorous reflection of the existing small mammal research. Small mammals have been the focal point of numerous public and management perspectives, sometimes leading to extended legal debate over wildlife management practices in an anthropogenic world (see work on Zapus in Colorado, prairie dogs, water shrews in Canada, and many other examples). I would think a better approach to this would be for the authors to actually assess the existing literature and summarize some of the prominent cases where small mammals have been an important consideration

L115 - citation is not consistent in text with references cited.

L128 - what does midden cruising survey mean?

L130 - who is RA?

L141 - Betula papyrifera

L152 - need to define old growth

L162 - most recent taxonomic revisions from Hope et al. 2016. Recognized by Natureserv and Illustrated Checklist of the Mammals of the World (C. J. Burgin, D. E. Wilson, R. A. Mittermeier, A. B. Rylands, T. E. Lacher, and W. Sechrest, eds.). Lynx Ediciones, Barcelona.

L163 - correct taxonomy and unitalicize “and” P. glauca and P. mariana (already mentioned previously)

L168 - Koprowski et al have published extensively on midden counts and characteristics. Density estimates of squirrels have been extensively covered in the literature by Boutin, Krebs, and others from the long term Kluane site.

L184-188 - this section does not provide any information on how the authors actually estimated ecological niche.

L202 - mittens?

L356 - refer to relevant figure

L367 - correct text for reference

L370 - explain rank of app.

L389 and onwards - it was not clear that any of these predictors were discussed in the methods section

L411 - rephrase these sentences and remove all encompassing parentheses

L431 - the European red squirrel is not closely related.

L441-443 - This set of statements is presumptive. Maybe it’s because trees nearer the lake are not growing on permafrost and therefore are more productive for cones.. There are many reasons why this observation might be the case, but jumping to the conclusion that correlation is causation is not appropriate. Also, given that red squirrels prefer high density canopy, again it is not surprising that they do not build middens near to trails where there is no canopy, and this may have no direct relationship to the presence of humans.

L475 - be consistent throughout the manuscript using decimal points or commas. Preferably points.

L480 - This sentence is unclear, throughout the manuscript refrain from using casual abbreviations for general words

L487 - So, this actually indicates that squirrel densities benefit from human proximity? This is counter to the central argument brought forth by the authors.

---

## Round 0.2 · Minor Revisions

We have now received feedback from two reviewers on your revision.
One of the reviewers requested which some additional to your manuscript, which we would like you to consider and address as appropriate.

Reviewer 1 ·

Basic reporting

No comment

Experimental design

No comment

Validity of the findings

No comment

Additional comments

No comment

Reviewer 2 ·

Basic reporting

Good. Seems like some of the figures could be moved to a supplement.

Experimental design

Good

Validity of the findings

Good

Additional comments

PEERJ-47121-v1

In their study “High-resolution prediction of American red squirrel in Interior Alaska: A role models for conservation using open access data, machine learning, GIS and LIDAR” the authors present a new combined predictive framework for assessing distribution and abundance of red squirrels in Alaska as related to human occupation. Fundamentally, this study provides a robust method towards the authors goals, and I appreciate that they addressed adequately all of the previous reviewer comments.

The major issues I still have with this article is the layout, sentence structure and content of the abstract and introduction which seem to have had less revision that the rest of the manuscript, maintaining some presumptive and broad brush statements as outlined by minor comments below. Additionally, the authors seek to provide a standardized applied methodology for global adoption by the scientific community, however, this set of methods relies heavily on identification and quantification of red squirrel middens. As such, I’m not sure how applicable these methods are for other species. I feel like they could be, but the authors have made no attempt to extrapolate how they might be employed for other species, to “be effective for being used by a global audience” as stated in the abstract. Red squirrels are only located in North America. So, it would I think be highly beneficial if the authors provided some examples of other species that produced diagnosable food caches for instance that may be assessed in a similar manner, or some other structural element of other species’ life histories that may lend to these methods being effective for the broader scientific community.

Minor comments:

L21 - Again, unless the authors are explicitly assessing evolutionary adaptation, this is not the most rigorous term to use. Perhaps “interacting” or “habituating”

L22 - why not state that squirrels build middens of cone debris to cache cone food resources and situate these middens central to their territories (if that is what the authors mean). This would provide more relevant information.

L47 - I do not think there is any direct evidence that avoidance is due to fear. In general the authors need to more carefully phrase their sentences to be scientifically objective and avoid making statements that weren’t explicitly tested or have been cited from the primary literature. Just state avoidance of human disturbance.

L74 - ubiquitous within North American boreal forests, or “within Alaskan forests”. Please check that all sentences are accurate. Given this is the start of a new paragraph, it requires more detail.

L79-81 - this statement should be cited. I am not sure how the authors can state this. If there is no literature published in journals, there may well still be government reports.

L96 - sentence structure.

L102 - sentence doesn’t make sense.

L374=376 - So, how do the authors discriminate between high density understory and high disturbance? It seems like these are opposite observations leading to the same predictive outcome, which would suggest ambiguity for predicting cause and effect of squirrel distributions. If “low trees” is the same as “low canopy height”, then the authors should use only one consistently throughout.

L401 - presumably tree height is also correlated to the maturity of trees, where no cones would be produced by trees under a certain size? As in, it may not be height per se, but age of tree.

---

## Round 0.3 · accepted · Accept

Thank you very much for addressing all the reviewer comments in your revised submission. I am satisfied with your responses and am happy to recommend the publication of your manuscript at this point.